# Antimicrobial-resistant pathogens on the plate: A semi-quantitative hygiene risk evaluation of raw beef consumption in Ethiopia within a One Health context

**Daniel Teshome Gebeyehu**[1,2*], **Temechew Munaw Abebe**[3], **Ayalew Negash Shiferaw**[2,4], **Md Shahidul Islam**[2], **Gashaw Enbiyale Kasse**[2,4]

**1** School of Veterinary Medicine, Wollo University, Dessie, Ethiopia, **2** School of Health, University of New England, Armiale, New South Wales, Australia, **3** College of Medicine and Health, Injibara University, Injibara, Ethiopia, **4** College of Veterinary Medicine and Animal Science, University of Gondar, Gondar, Ethiopia

* daniel.teshome@wu.edu.et

## Abstract

### Background

While meat is a valuable source of nutrition, it can also serve as a vehicle for infectious and non-infectious diseases—especially when consumed raw. This risk becomes more serious when the meat harbors antimicrobial-resistant bacteria, often referred to as superbugs. This study assessed the health risks faced by consumers of raw beef in relation to infections from common enteric superbugs commonly transmitted through meat.

### Methods

The investigation focused on detecting the prevalence of three key meat-borne bacteria—*Salmonella*, *Shigella*, and *Escherichia coli*—and evaluating their resistance to commonly used antibiotics. Bacterial identification was performed using culture techniques and biochemical tests, while antibiotic susceptibility was assessed via the disc diffusion method. The study also examined food safety practices at restaurants that serve raw beef, including hygiene measures and whether the meat underwent any thermal treatment. Health risk estimation was based on both likelihood factors (bacterial contamination and hygiene practices) and consequence factors (resistance to antibiotic treatments).

### Results

Meat samples showed an overall bacterial contamination rate of 45.2%, with *E. coli* being the most prevalent (62.8%), followed by *Shigella* (32.7%) and *Salmonella* (4.4%). Alarmingly, 26.8% of the *E. coli* isolates were identified as the highly

**Data availability statement:** All data generated or analysed during this study are included in this published article and its supplementary information files.

**Funding:** This study was funded by Wollo University (Funding No. 15676), which provided financial support in the form of per diem allowances for data collectors and covered the costs of laboratory reagents and consumable materials. There was no agreement between the authors and the funder regarding whether or not to publish the findings, and the funder had no role in study design, data collection and analysis, decision to publish, or preparation of the manuscript.

**Competing interests:** The authors have declared that no competing interests exist.

pathogenic *E. coli O157:H7* strain. None of the restaurants applied any form of heat treatment (cooking or chilling) to the raw beef before serving, and 76% had poor hygiene practices based on bacterial contamination findings. All isolates were fully resistant to amoxicillin, and *Salmonella* strains also showed complete resistance to erythromycin. However, all bacteria remained 100% sensitive to chloramphenicol. Most isolates demonstrated multidrug resistance to commonly used antibiotics including ciprofloxacin, doxycycline, erythromycin, and streptomycin. The estimated health risk to raw beef consumers was high, primarily due to significant exposure levels and limited effectiveness of post-exposure treatment options.

## Conclusion

The combined effects of high bacterial contamination, poor hygiene, lack of heat treatment, and widespread antibiotic resistance pose a potential hygiene-associated risk to raw beef consumers. Reducing this risk requires prompt action, including raising public awareness, enforcing strict meat hygiene standards, and implementing strategies to curb antimicrobial resistance.

---

## Introduction

Meat remains a cornerstone of human nutrition, offering a rich source of protein, essential amino acids, vitamins (notably B12), iron, zinc, and other micronutrients crucial for sustaining physical and cognitive health [1,2]. However, while it plays a vital role in promoting well-being, meat—especially when consumed raw—can serve as a potent vehicle [3,4] for transmitting foodborne pathogens [5] and non-communicable health risks [6]. With the escalating global crisis of antimicrobial resistance (AMR), the health risks associated with the consumption of raw meat have become even more alarming [7–9]. Pathogenic bacteria such as *Salmonella*, *Shigella*, and *Escherichia coli* (including *E. coli O157:H7*) that are resistant to multiple antibiotics—often referred to as "superbugs"—can severely limit treatment options, leading to prolonged illness, complications, or even death [10,11].

Heat-based processing techniques like cooking, boiling, or roasting is essential for eliminating harmful microorganisms in meat [12]. However, these safety barriers are bypassed when meat is consumed raw—significantly increasing the risk of infection [13–15]. In Ethiopia, the tradition of raw beef consumption (*kurt* or *kitfo*) is deeply rooted in cultural heritage [16]. Historically, this practice is said to have originated during times of armed conflict, when soldiers sought to avoid smoke that might reveal their positions. Today, it remains a popular delicacy, especially during festive occasions and communal gatherings [16].

Despite its popularity, the widespread practice of consuming raw beef in Ethiopia is fraught with public health risks. Most slaughtering practices outside formal municipal abattoirs are carried out in open spaces or poorly regulated settings, often lacking essential sanitary infrastructure [17,18]. Even within authorized abattoirs and

retail meat shops, hygiene standards are inconsistently applied, resulting in a high potential for microbial contamination [17,19,20]. It was reported that the contamination rates of 8.6% for *Salmonella* and 7.1% for *E. coli O157:H7* in meat samples from selected abattoirs of Addis Ababa [21]. Notably, 57.14% of *E. coli O157:H7* isolates from raw beef in Ethiopia exhibited multidrug resistance [13], underscoring the urgency of addressing AMR threats in the food chain [22].

Unlike high-income countries such as Australia and New Zealand [23], the United States [24], and members of the European Union [25]—where meat production is governed by stringent safety regulations and coordinated monitoring systems—Ethiopia faces systemic gaps in meat safety governance [17,21,26–30]. Regulatory responsibilities for meat safety are fragmented between the animal health and public health sectors, with limited intersectoral collaboration or consistent implementation of food safety standards. Although the recent establishment of Ethiopia's national One Health steering committee marks a step in the right direction, the practical integration of veterinary and public health efforts remains weak [31].

In addition to structural regulatory gaps, public awareness of foodborne risks associated with raw meat consumption remains low. Consumer habits, lack of risk communication, and the informal nature of meat markets further compound the problem. Existing studies have documented microbial contamination of meat in various Ethiopian settings, yet few have quantified the specific health risks to raw beef consumers posed by antibiotic-resistant pathogens. This critical gap limits evidence-based policy interventions and hinders the development of culturally sensitive, risk-reducing public health strategies. AMR in key foodborne enteric pathogens such as *Escherichia coli*, *Salmonella*, and *Shigella* poses a significant public health threat, as rising resistance levels have been linked to treatment failure, particularly in settings where therapeutic options are limited [32]. These organisms are among the most common Gram-negative enteric pathogens linked to foodborne illness worldwide and are designated by the World Health Organization as high-priority antimicrobial-resistant pathogens due to their clinical significance and frequent association with contaminated foods, especially raw meat [33].

In this context, the present study aims to assess potential health risks associated with raw beef consumption in Ethiopia, focusing specifically on the prevalence and antimicrobial resistance patterns of *Salmonella*, *Shigella*, and *E. coli* (including the *O157:H7* serotype). This study integrates AMR profiling, microbial contamination, and field-level hygiene observations across the raw beef supply chain, an area underreported in Ethiopia to generate semi-quantitative insights that can inform national food safety policies, raise public awareness, and support coordinated One Health interventions addressing antimicrobial resistance in the food chain.

## Materials and methods

### Study area

This study was conducted in selected towns from the South Wollo and Oromia zones of the Amhara Regional State in northeastern Ethiopia. Specifically, samples were collected from five towns: Dessie, Kombolcha, and Wereilu in South Wollo, and Kemise and Bati in Oromia. According to the Ethiopian Statistical Service [34], the projected population of the South Wollo zone is estimated at approximately 3,435,377, while the Oromia zone has around 619,868 inhabitants. These areas were selected due to their high prevalence of raw beef consumption and established meat markets. The towns are located at distances of 401 km (Dessie) and 327 km (Oromia zone towns) from Ethiopia's capital city, Addis Ababa, making them accessible for logistics and laboratory transport. These regions exhibit mixed urban-rural characteristics and variable sanitation infrastructure, which presents a suitable context for assessing hygiene practices and associated food safety risks.

### Study population and sampling

The target population comprised licensed restaurants known for serving raw beef (commonly known as "kitfo", "gored gored" and "kurt") within the five selected towns. From an estimated 200 licensed food establishments across the five study towns, a total of 50 restaurants were selected using a stratified random sampling approach. Sampling frames (restaurant lists) were obtained from each city's municipal office. To ensure proportional representation, the number of

restaurants selected from each town reflected its relative share of establishments: 23 from Dessie, 7 from Kombolcha, 9 from Kemise, 5 from Bati, and 6 from Wereilu. Within each town, restaurants were randomly selected using a lottery method. While the sampling was random, stratification ensured inclusion of restaurants from areas with varied customer traffic and infrastructure standards, allowing a realistic representation of hygiene conditions.

At each selected restaurant, a one-time sampling event was conducted during operational hours. Five swab samples were collected from key contact surfaces, meat cutting boards, knives, pans, serving tables, and food handlers' hands-yielding a total of 250 swab samples. Sampling was conducted between January and June 2023, with each establishment visited once during the study period. In addition to microbiological sampling, observational assessments were conducted to evaluate hygiene practices, meat handling protocols, and environmental sanitation.

## Study design

A community-based cross-sectional study design was employed between January 2023 and June 2023. This design was chosen to enable the evaluation of current food safety practices and microbial contamination levels at a single point in time across various towns. It also facilitated comparative analysis across different geographical locations and socio-economic contexts. This approach aligns with previous food safety research that aims to identify and quantify contamination risks without requiring long-term observation [35].

## Ethics approval

Approval to conduct this study was obtained from the Institutional Review Board (IRB) of Wollo University (Approval No. WU/15676/N05/13). Prior to data collection, restaurant owners were fully informed about the objectives and procedures of the study. Written informed consent was obtained from each participating restaurant owner before initiating the research in their establishments.

For the participating meat cutters (butchers), the study procedures were explained in detail, including the purpose of hand swab sampling and assurances of confidentiality. Verbal informed consent was obtained from each butcher prior to participation, as most participants were more comfortable providing oral consent. This verbal consent process was documented by the research team through signed witness statements by the data collectors for each participating butcher. No minors were involved in this study.

To maintain confidentiality, the identities of all sampled restaurants and participants were anonymized. Each butcher was assigned a numerical ID, and no personal identifiers or real names were recorded at any stage of the study.

## Data collection and management

A multi-tiered data collection strategy was implemented to capture both quantitative microbial contamination levels and qualitative hygiene assessments. All registered restaurants were first listed in collaboration with local food safety authorities. The 50 selected restaurants were then visited by trained field investigators equipped with sterile sampling materials and standardized checklists.

Samples were collected using sterile swabs soaked in 10 ml of peptone saline solution. For each sample type, specific protocols were followed: knives and cutting boards were swabbed over a defined 100 cm$^2$ area, pans were sampled from top, middle, and bottom layers, and meat surfaces were sampled from four anatomical locations (rump, flank, chest/sternum, and foreleg), based on Mullen [36]. Samples were immediately stored in sterile tubes, placed in iceboxes, and transported within 24 hours to the microbiology laboratory at Wollo University. Suspected isolates were preserved at −20°C for subsequent testing.

In tandem, observational checklists based on CDC food safety guidelines [37] were used to assess hygiene indicators such as equipment cleanliness, hand hygiene, use of personal protective equipment (PPE), toilet proximity, and sink availability. Hygiene scores were later compiled and quantified to estimate cumulative risk.

## Hazard identification

Hazard identification focused on isolating and identifying pathogenic bacteria known to be associated with raw beef contamination, specifically *Salmonella spp.*, *Shigella spp.*, *Escherichia coli* O157:H7, and other *E. coli* strains. Isolation protocols were adapted from the U.S. Food and Drug Administration's Bacteriological Analytical Manual [38]. Pre-enrichment involved incubating 1 ml of each sample in 9 ml buffered peptone water at 37°C for 18 hours. For selective enrichment, 0.1 ml of this suspension was transferred to Rappaport-Vassiliadis broth and incubated at 42°C for 24 hours.

Samples were then streaked onto MacConkey Agar to identify lactose-fermenting and non-fermenting colonies. Suspect colonies were further sub-cultured on *Salmonella-Shigella* agar and XLD agar, and biochemical confirmation tests (including Indole, Methyl Red, Citrate, TSI, Urease, and sugar fermentation) were conducted as per Garrity [39].

For *E. coli* O157:H7, differentiation involved initial culturing on MacConkey Agar followed by transfer to Eosin Methylene Blue (EMB) Agar and Sorbitol MacConkey (SMAC) Agar. Colonies showing a green metallic sheen on EMB and non-sorbitol fermentation on SMAC were subjected to additional confirmation. Non-sorbitol fermenting colonies appeared colourless, then were further cultured on nutrient agar and subjected to biochemical testing following Garrity [39].

Owing to resource and equipment constraints both in the study laboratory and nationally, pathogen identification was performed using conventional culture and biochemical methods, without PCR or serotyping.

## Antibiotic resistance testing

Antimicrobial susceptibility testing was performed using the disk diffusion method as per Clinical and Laboratory Standards Institute [40] protocols. Each isolate was inoculated into saline to achieve a 0.5 McFarland turbidity standard. A sterile swab was used to spread the culture onto Mueller-Hinton agar plates. Antibiotic discs of six commonly used antimicrobials, including erythromycin (15 μg), doxycycline (30 μg), chloramphenicol (30 μg), amoxicillin (2 μg), streptomycin (10 μg), and ciprofloxacin (30 μg) were applied.

The selected antibiotics still in local use for human and veterinary treatment, such as amoxicillin and erythromycin, to monitor retained or emerging resistance patterns relevant to the Ethiopian context. Plates were incubated at 37°C for 18 hours and zones of inhibition were measured with digital callipers. Results were interpreted based on CLSI-defined thresholds for categorizing isolates as Sensitive, Intermediate, or Resistant. Isolates lacking inhibition zones were classified as fully resistant, indicating potential public health threats. In addition to assessing resistance levels of each isolate to individual antibiotics, this study also evaluated multidrug resistance (MDR), defined as resistance to at least one agent in three or more antimicrobial classes. MDR poses a serious challenge to treatment efforts, as it significantly limits therapeutic options and complicates infection management, particularly in low-resource settings where alternative drugs are often unavailable.

Antimicrobial susceptibility testing quality control was conducted using *E. coli ATCC 25922* and *S. aureus ATCC 25923*, in accordance with CLSI guidelines [40]. The antibiotics included in this study were chosen based on their availability, routine use in local veterinary and human medicine in in the study area, and relevance for resistance surveillance. Breakpoints as indicated in CLSI were applied where applicable; for antibiotics without specific breakpoints for the tested organisms, interpretive thresholds from relevant literature were used.

Antimicrobial susceptibility testing was performed using the Kirby–Bauer disk diffusion method following Clinical and Laboratory Standards Institute (CLSI) guidelines [40]. Each isolate was suspended in sterile saline and adjusted to a 0.5 McFarland turbidity standard. The suspension was uniformly spread onto Mueller–Hinton agar plates using a sterile swab. Antibiotic discs of six commonly used antimicrobials were applied: erythromycin (15 μg), doxycycline (30 μg), chloramphenicol (30 μg), amoxicillin (2 μg), streptomycin (10 μg), and ciprofloxacin (30 μg).

The antibiotics were selected to reflect antimicrobials commonly used in human and veterinary practice in the study region and to support AMR surveillance in the food chain. he WHO Advisory Group on Integrated Surveillance

of Antimicrobial Resistance (WHO AGISAR) recommends that antimicrobial panels for surveillance studies should consider local antimicrobial use patterns and public health priorities and recognizes *Escherichia coli* as an important sentinel organism for monitoring antimicrobial selection pressure and the dissemination of resistance determinants within food production systems [41].

Previous studies have similarly included amoxicillin and erythromycin in susceptibility testing panels for Gram-negative enteric bacteria such as *E. coli*, *Salmonella*, and *Shigella* in both clinical and food-chain investigations. For example, Kazemnia et al. [42] reported antimicrobial susceptibility testing of *E. coli* isolates including amoxicillin and erythromycin, demonstrating high resistance levels to both agents. Fatima et al. [43] and Kim et al. [44] investigated antimicrobial resistance among foodborne *Salmonella* isolates and reported resistance patterns including erythromycin. Likewise, Ayele et al. [45] evaluated antimicrobial susceptibility among *Shigella* isolates in Ethiopia and included both amoxicillin and erythromycin in the testing panel. These studies demonstrate that inclusion of these antibiotics in susceptibility testing panels for Enterobacterales has been previously applied in AMR surveillance and epidemiological studies.

Plates were incubated at 37 °C for 18 hours, and zones of inhibition were measured using digital callipers. Results were interpreted according to CLSI breakpoint criteria for Enterobacterales where available [46]. For organism, antibiotic combinations lacking validated CLSI or EUCAST clinical breakpoints, results were interpreted cautiously and considered primarily for surveillance purposes rather than direct therapeutic guidance.

Multidrug resistance (MDR) was defined as resistance to at least one antimicrobial agent in three or more antimicrobial classes [47]. MDR is a major public health concern because it substantially limits treatment options and complicates infection management. For MDR analysis, intrinsic resistance traits were not considered when classifying isolates as MDR in order to avoid overestimation of resistance levels.

Quality control for antimicrobial susceptibility testing was performed using *Escherichia coli* ATCC 25922 and *Staphylococcus aureus* ATCC 25923 in accordance with CLSI standards [40].

## Health risk assessment

A structured, semi-quantitative hygiene risk evaluation was conducted to assess the potential public health risks posed by antibiotic-resistant pathogens in raw beef. The approach was informed by principles from Codex Alimentarius Commission and WHO food safety risk analysis frameworks [48,49], but did not follow a formal quantitative modelling process. Instead, this study focused on assessing pathogen prevalence, contamination pathways, resistance profiles, and their potential implications for public health.

Risk was conceptually examined in two observational stages: pre-exposure and post-exposure. Pre-exposure evaluation involved the assessment of hygiene practices and sanitation at meat handling and serving points, including equipment cleanliness, PPE usage, and general environmental conditions. Post-exposure considerations included the implications of local antibiotic resistance patterns for potential treatment outcomes.

Hygiene indicators were scored using a binary (compliant/non-compliant) approach and averaged to reflect adherence to baseline standards. A benchmark toilet distance of 10 meters was applied based on WHO recommendations [4]. Hygiene control gaps were calculated from cumulative non-compliance across categories such as surface hygiene, PPE usage, and infrastructure quality. These observational scores were combined with AMR findings to characterize relative consumer risk in the absence of formal probability-based modelling.

## Data analysis

The collected data underwent both quantitative and qualitative analyses, structured according to international food safety risk assessment frameworks provided by Codex Alimentarius and the World Health Organization [35,49]. After bacterial identification and antibiotic susceptibility testing, the analysis proceeded to a comprehensive risk assessment

phase that incorporated hazard characterization, exposure assessment, consequence analysis, and final risk estimation.

Initially, bacterial isolates from meat surfaces, food-contact utensils, and handlers' hands were characterized in terms of frequency of occurrence and resistance patterns. Descriptive statistics were used to determine the prevalence of each bacterial species, their distribution across towns, and the percentage of samples exhibiting antimicrobial resistance. Resistance levels were interpreted based on inhibition zone diameters as per the Clinical and Laboratory Standards Institute [40] guidelines, categorizing isolates as Sensitive, Intermediate, or Resistant.

The risk evaluation was conducted by deriving an observational composite score for each sample site using the expression: Health Risk = Likelihood of exposure × Severity of consequence. This formulation was adapted for practical use in field settings and is not based on a formally validated risk estimation model. The likelihood of exposure was approximated by assessing hygiene-related gaps, such as deficiencies in surface sanitation, PPE usage, infrastructure adequacy, and hand-washing compliance, each scored on a binary (compliant/non-compliant) scale. The cumulative percentage of non-compliance across these parameters was used to indicate relative exposure potential at each site. The binary scoring system was adapted from WHO hygiene checklists. Cumulative risk categories were based on total non-compliant items and not weighted by exposure probability or severity, limiting precision.

The severity of consequence was approximated using antimicrobial resistance profiles of isolated pathogens. Specifically, resistance rates for *Salmonella*, *Shigella*, and *E. coli O157:H7* were used as proxies to reflect potential treatment challenges. Based on the qualitative classification approach described in a study [50], sites were categorized into low, medium, or high relative risk tiers. While not predictive in a probabilistic sense, this matrix provides a practical field tool to contextualize hygiene and AMR findings within a public health risk framework.

All results were entered and processed using SPSS (version 28). Descriptive statistics—including frequency distributions, percentages, means, and standard deviations—were generated. Although the data collection design included hygiene observations and microbial testing, statistical testing for associations between hygiene practices and contamination rates was not performed due to limited subgroup sizes. Risk patterns were instead summarized using composite scores and categorical interpretation of hygiene and resistance data. Spatial variation in observed risk levels across towns was described to highlight potential public health concerns.

Finally, health risks were contextualized in terms of morbidity and mortality implications, informed by literature on food-borne illness burden. Infections with multi-drug-resistant organisms were considered to pose the most severe outcomes, particularly among vulnerable populations such as children, the elderly, and immunocompromised individuals [4,49,51]. Recommendations for intervention were formulated based on the magnitude and distribution of calculated risks, targeting areas with high hygiene gaps and AMR prevalence.

## Results

### Hazard identification assessment

This study focused on three common foodborne bacterial pathogens: *Salmonella*, *Shigella*, and *E. coli*. A total of 250 samples were collected from ready-to-eat raw beef, raw beef preparation equipment (including knives, tables, and pans), meat cutters' hands, and meat surfaces. Among these, 45.2% tested positive for the target pathogens—*Salmonella*, *Shigella*, *E. coli O157:H7*, and other *E. coli* strains (Table 1). The detailed general characteristics of the samples are provided in Supporting information (S1 Text Excel) and summarized in Table 1.

### Risk exposure and mitigation assessments

Common measures to reduce health risks associated with biological hazards in raw beef include temperature control (via cooking or cooling), hygienic practices (such as cleaning and disinfection), and post-exposure antibiotic treatment.

**Table 1. General information on the location and type of equipment that the bacterial hazards identified.**

| | Categories | Total samples (n = 250) | Positive samples (n = 113) | Type and number of bacterial hazards | | | |
|---|---|---|---|---|---|---|---|
| | | | | *Salmonella* | *Shigella* | *Other E. coli strains* | *E. coli O157:H7* |
| Location | Dessie | 115 | 45(39.82%) | 2 | 12 | 25 | 6 |
| | Kombolcha | 35 | 25(22.12%) | 0 | 9 | 10 | 6 |
| | Kemissie | 45 | 18(15.93%) | 1 | 4 | 10 | 3 |
| | Bati | 25 | 9(7.96%) | 1 | 0 | 4 | 4 |
| | Woreilu | 30 | 16(14.16%) | 1 | 12 | 3 | 0 |
| Equipment | Knifes | 50 | 23(20.35%) | 0 | 12 | 9 | 2 |
| | Pan | 50 | 28(24.78%) | 0 | 8 | 15 | 5 |
| | Tables | 50 | 24(21.24%) | 2 | 6 | 6 | 10 |
| | Hands | 50 | 19(16.81%) | 3 | 5 | 10 | 1 |
| | Meat surface | 50 | 19(16.81%) | 0 | 6 | 12 | 1 |
| Total positive isolates | | | 113(45.2%) | 5 (4.43%) | 37(32.74%) | 52(46.02%) | 19(16.81%) |

## Temperature mitigation assessment

During field observations, it was confirmed that the sampled raw beef was served without any heat treatment, as the samples were collected from restaurants offering raw beef dishes. Consequently, temperature-based control measures—like heating or chilling—were not implemented.

## Hygienic control assessment

The findings revealed significant shortcomings in hygiene across many establishments. Rough cutting boards were found in 62% of restaurants, and 34% had unclean washing baths. Additionally, 86% of raw beef-serving butchers were not using personal protective equipment (PPE). Unhygienic working environments were noted in 46% of the restaurants, while 38% had dining areas located less than 10 meters from the toilets. Furthermore, 44% of dining tables were unclean, and 76% of the restaurants tested positive for the target bacterial pathogens (Table 2). These statistics highlight a clear lack of effective hygienic risk mitigation and control practices.

**Table 2. The restaurants serving raw beef' hygienic management practice.**

| Assessment criteria | Grading by the researcher | Numbers | Percent |
|---|---|---|---|
| Cutting board smoothness (easy for cleaning) | Smooth | 19 | 38% |
| | Rough | 31 | 62% |
| Washing bath cleanness | Clean | 33 | 66% |
| | Not clean | 17 | 34% |
| Butchers' PPE using practice | Use | 7 | 14% |
| | Not use | 43 | 86% |
| General restaurant hygiene | Poor | 23 | 46% |
| | Medium | 11 | 22% |
| | Good | 16 | 32% |
| Toilet distance from meat cutting and dining areas | Far (>10m) | 31 | 62% |
| | Near (<10m) | 19 | 38% |
| Dining table cleanness | Clean | 28 | 56% |
| | Not clean | 22 | 44% |
| Prevalence of bacterial hazards | Positive | 38 | 76% |
| | Negative | 12 | 24% |

The high prevalence of bacterial contamination on meat surfaces and equipment, combined with the absence of pre-consumption temperature treatment and poor hygienic practices, significantly increases the risk of infection for raw beef consumers. Pathogens of particular concern include *Salmonella*, *Shigella*, and *E. coli*.

## Antibiotic mitigation assessment

Antibiotic treatment is one of the primary interventions used to manage post-infection outcomes such as morbidity and mortality. However, widespread use of antibiotics contributes to the growing issue of antimicrobial resistance (AMR). To evaluate the effectiveness of antibiotic treatments against bacteria originating from raw beef, antimicrobial susceptibility testing was conducted on bacterial isolates collected from raw beef-serving restaurants. Six commonly used antibiotics were tested in this investigation.

The results showed that all (100%) bacterial isolates—including *Salmonella*, *Shigella*, *E. coli*, and *E. coli* O157:H7— were sensitive to chloramphenicol. In contrast, sensitivity to streptomycin varied: 40% of *Salmonella*, 57% of *Shigella*, 53% of other *E. coli* strains, and 58% of *E. coli O157:H7* showed susceptibility (Fig 1).

A significant proportion (60%) of *E. coli* isolates—excluding *E. coli* O157:H7—exhibited intermediate sensitivity to ciprofloxacin. In contrast, all other bacterial isolates demonstrated less than 50% intermediate sensitivity to ciprofloxacin, doxycycline, erythromycin, and streptomycin (Fig 2).

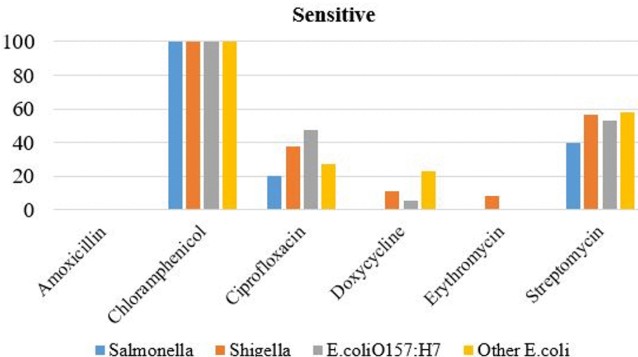

**Fig 1. Antibiotic sensitivity level of bacteria isolated from raw beefs and equipment.**

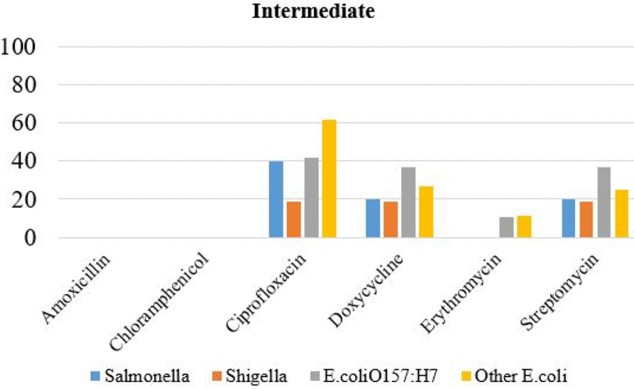

**Fig 2. Intermediate antibiotic sensitivity level of bacteria isolated from raw beefs and equipment.**

All bacterial hazards were completely resistant to amoxicillin, and all *Salmonella* isolates showed 100% resistance to erythromycin (Fig 3). Additionally, 80% of *Salmonella*, 70% of *Shigella*, and 58% of *E. coli O157:H7* isolates were resistant to doxycycline. High resistance to erythromycin was also observed in 92% of *Shigella*, 89.5% of *E. coli* O157:H7, and 88.5% of unidentified *E. coli* strains (Fig 3).

Except for chloramphenicol, all bacterial isolates exhibited multidrug resistance (MDR) to the antibiotics tested – amoxicillin, ciprofloxacin, doxycycline, erythromycin, and streptomycin. Notably, *Salmonella* isolates showed higher resistance to doxycycline, erythromycin, and streptomycin compared to *Shigella* and *E. coli*. Moreover, *Shigella* isolates demonstrated greater resistance than *E. coli* to ciprofloxacin, doxycycline, erythromycin, and streptomycin.

The most consistent inhibition zone measurements – indicated by the lowest variability – were observed with amoxicillin (SD = 0). In contrast, erythromycin exhibited the highest variability (SD = 3.46), followed by doxycycline (SD = 2.87) (Table 3). Detailed variability data for inhibition zones across all three antibiotic resistance categories (sensitive, intermediate, and resistant) are presented in Table 3. The detailed data used to calculate means, standard deviations, standard errors, and overall resistance percentages are available in supporting information (S2 Excel).

### Health risk assessment in relation to pre and post exposure assessment results

The assessment revealed a substantial hygienic control gap among restaurants serving raw beef, with an average shortfall of 52% (Table 4). As the raw beef in these establishments was consumed without any form of thermal processing— neither cooking nor chilling—the temperature control gap was identified as 100% (Table 4). The prevalence rates of key bacterial pathogens, including *Shigella spp.*, *Salmonella spp.*, and *Escherichia coli*, were derived from the microbiological surveillance data summarized in Table 2.

By integrating three primary likelihood factors—the complete absence of temperature control, high levels of bacterial contamination, and considerable deficiencies in hygienic practices—alongside the severity factor, which accounts for antimicrobial resistance (AMR) and multidrug resistance (MDR) to commonly used antibiotics, the overall public health risk linked to raw beef consumption in the evaluated restaurants was classified as high. This categorization exceeds the defined thresholds for both low and medium risk levels, as detailed in Table 4.

The severity of the associated health risks is further reinforced by internationally recognized antimicrobial resistance (AMR) indicators. According to the World Health Organization [33], resistance rates of 42% to third-generation cephalosporins in *Escherichia coli* and 35% methicillin resistance in Staphylococcus aureus (MRSA) are classified as high, representing a significant global public health threat. Moreover, *Salmonella* spp., *Shigella* spp., and *E. coli*—the three key enteric pathogens commonly linked to raw meat consumption—are all designated as high-priority AMR pathogens by WHO [32]. When these alarming resistance patterns are considered alongside widespread gaps in temperature

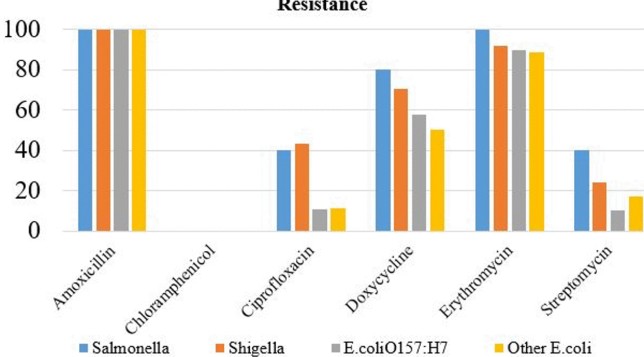

**Fig 3. Antibiotic resistance level of bacteria isolated from raw beefs and equipment.**

**Table 3. Variability and consistency of the inhibition zones using means, standard deviation, and standard error.**

| Isolated Hazards (n = 113) | Antibiotics | Sensitive | | | Intermediate | | | Resistant | | |
|---|---|---|---|---|---|---|---|---|---|---|
| | | Mean | S. D | S. E | Mean | S. D | S. E | Mean | S. D | S. E |
| *Salmonella* | Amoxicillin | – | – | – | – | – | – | 6 | 0 | 0 |
| | Chloramphenicol | 24.6 | 4.56 | 2.04 | – | – | – | – | – | – |
| | Ciprofloxacin | 23 | – | – | 20 | 0 | 0 | 15.5 | 2.12 | 1.5 |
| | Doxycycline | – | – | – | 13 | – | – | 7.75 | 2.87 | 1.45 |
| | Erythromycin | – | – | – | – | – | – | 8 | 3.46 | 1.55 |
| | Streptomycin | 16 | 1.41 | 1.00 | 14 | – | – | 9 | 1.41 | 1.00 |
| *Shigella* | Amoxicillin | – | – | – | – | – | – | 6 | 0 | 0 |
| | Chloramphenicol | 28.5 | 3.31 | 0.55 | – | – | | – | – | – |
| | Ciprofloxacin | 28.6 | 3.41 | 0.91 | 20.4 | 0.79 | 0.29 | 15 | 2.53 | 0.63 |
| | Doxycycline | 17.5 | 1.73 | 0.87 | 13 | 1.63 | 0.62 | 7 | 1.37 | 0.27 |
| | Erythromycin | 24.3 | 1.16 | 0.67 | – | – | – | 6.7 | 1.17 | 0.20 |
| | Streptomycin | 16 | 1.62 | 0.35 | 13 | 0.69 | 0.26 | 10 | 0.84 | 0.29 |
| Other *E. coli* species | Amoxicillin | – | – | – | – | – | – | 6 | 0 | 0 |
| | Chloramphenicol | 27.3 | 3.91 | 0.54 | – | – | – | – | – | – |
| | Ciprofloxacin | 25 | 1.14 | 0.31 | 20 | 0.80 | 0.14 | 16 | 2.64 | 1.09 |
| | Doxycycline | 15.7 | 1.83 | 0.53 | 12.3 | 0.83 | 0.22 | 7.9 | 1.47 | 0.29 |
| | Erythromycin | – | – | – | 16.3 | 2.12 | 0.71 | 12.9 | 2.87 | 0.42 |
| | Streptomycin | 15.3 | 0.60 | 0.11 | 12.8 | 0.93 | 0.26 | 8.2 | 1.30 | 0.43 |
| *E. coli O157:H7* | Amoxicillin | – | – | – | – | – | – | 6 | 0 | 0 |
| | Chloramphenicol | 27.4 | 2.95 | 0.68 | – | – | – | – | – | – |
| | Ciprofloxacin | 25.6 | 1.88 | 0.63 | 20.3 | 0.71 | 0.25 | 17 | – | – |
| | Doxycycline | 17 | 2.83 | 2.00 | 14 | 1.07 | 0.40 | 10 | 2.05 | 0.62 |
| | Erythromycin | – | – | – | 12.5 | 2.12 | 1.50 | 8 | 1.69 | 0.41 |
| | Streptomycin | 15..6 | 1.27 | 0.40 | 12.7 | 0.76 | 0.29 | 9.5 | 0.71 | 0.50 |

S.D = Standard deviation, S.E = Standard error, n = total number of positive samples.

regulation, sanitary practices, and hygienic handling throughout the meat supply chain, the risk of consumer exposure to multi-drug resistant (MDR) bacteria through raw beef becomes markedly elevated.

## Discussion

The present study identified a disturbingly high prevalence (45.2%) of bacterial contamination, particularly common enteric food contaminants *Escherichia coli* (*E. coli*), *Shigella*, and *Salmonella* [52, 53] in restaurants serving raw beef. Among these, *E. coli* was the most frequently isolated pathogen (62.83%), followed by *Shigella* (32.74%) and *Salmonella* (4.43%). The elevated prevalence of *Shigella* and *E. coli O157:H7* should be interpreted with caution given the absence of molecular confirmation. These findings highlight a significant public health concern, as consumption of contaminated raw beef can result in severe foodborne illnesses with both acute and long-term health consequences [54].

The predominance of *E. coli* aligns with other studies conducted in similar settings. For example, raw meat samples in Ghana showed even higher *E. coli* contamination (84%) [55], while similar trends were observed in Nigeria (64.1%) [56] and Sri Lanka [57] (66.8%) for raw chicken meat. Particularly concerning is the presence of Shiga toxin-producing *E. coli* (STEC) *O157:H7*, a highly virulent strain [51] that accounted for 16.81% of all *E. coli* isolates in the current study. This prevalence is notably higher than previously reported rates in Addis Ababa (3.64%) [13], Arsi (2.1%) [58], and other Ethiopian regions (0.54%) [59]. Such variation could be attributed to differences in hygiene practices, public

**Table 4. Health risk estimations of raw beef consumers using pre and post exposure mitigation gaps.**

| Likelihood factors | Gaps in the control measures | MDR of the bacterial hazards | | Health risks estimations |
|---|---|---|---|---|
| | | Isolates | Cumulative% | |
| Cumulative temperature control (cooking/cooling) of the restaurants | 100% (1) | *Salmonella* | 72% MDR | High |
| | | *Shigella* | 65% MDR | High |
| | | *E. coliO157:H7* | 54% MDR | High |
| | | *Other E. coli Species* | 53% MDR | High |
| Cumulative prevalence of the bacterial isolates | 76% (0.76) | *Salmonella* | 72% MDR | High |
| | | *Shigella* | 65% MDR | High |
| | | *E. coliO157:H7* | 54% MDR | High |
| | | *Other E. coli Species* | 53% MDR | High |
| Cumulative hygienic control of the restaurants | 52% (0.52) | *Salmonella* | 72% MDR | High |
| | | *Shigella* | 65% MDR | High |
| | | *E. coliO157:H7* | 54% MDR | High |
| | | *Other E. coli Species* | 53% MDR | High |

health infrastructure, and awareness levels among food handlers and consumers [60]. While most *E. coli* strains are non-pathogenic and serve as indicators of faecal contamination, the isolation of *E. coli O157:H7*, a known enterohemorrhagic strain, highlights potential public health hazards associated with raw beef consumption.

*Shigella* is recognized as the second-leading cause of diarrheal-related deaths globally, contributing to over 212,000 deaths in 2016 alone [61]. Its substantial prevalence (32.74%) in raw beef-serving establishments in the current study presents a significant public health threat. By contrast, lower prevalence rates have been reported in India (18.7%) [62], Ethiopia (10.5%) [63], Iran (4.44%) [64], and other parts of Ethiopia (3%) [65]. These differences may reflect varying levels of food safety training, regulatory enforcement, and sanitation standards across regions.

Several biological and ecological factors may explain the relatively high prevalence of enteric bacterial contamination observed in raw beef samples. Slaughterhouse environments and meat handling practices play a critical role in the transmission of enteric pathogens from animal intestines to carcasses during evisceration. Cross-contamination may occur through contaminated equipment, workers' hands, and processing surfaces. In addition, traditional meat processing environments in many developing regions often lack strict hygiene control, which can facilitate bacterial transmission along the meat supply chain.

Supply chain conditions may also contribute to bacterial contamination. Inadequate refrigeration during transportation and storage can allow rapid bacterial proliferation. Interruptions in the cold chain, particularly in warm climates, may permit growth of enteric pathogens such as *E. coli*, *Salmonella*, and *Shigella*. These conditions highlight the importance of temperature control and improved food safety management systems to reduce contamination risks.

Although *Salmonella* was detected at a comparatively lower rate (4.43%), it remains a critical pathogen due to its high global burden. Non-typhoidal *Salmonella* caused an estimated 535,000 illnesses and 77,500 deaths worldwide in 2017 [66]. The observed prevalence, while lower than reported in Gondar [67] (5.5%) and Italy [68] (5.84%), underscores the persistent risk posed by inadequate food safety measures.

Several critical hygiene deficiencies were identified as contributing factors [69] to microbial contamination. For instance, 62% of the surveyed restaurants used rough-surfaced cutting boards, which are known to retain microbial residues [70] and are difficult to sanitize effectively [70]. A study by Yang, Kendall (70) found that *Listeria monocytogenes* adhered more strongly to rough surfaces than to smooth ones, making decontamination more challenging. Additionally, 86% of raw beef-serving establishments lacked the use of personal protective equipment (PPE), increasing the likelihood of cross-contamination between food handlers and consumers. Comparable findings were reported in the U.S., where 60% of food handlers failed to use PPE and 53% did not monitor temperature control during meat processing [71].

The proximity of toilets to dining areas in 38% of the restaurants—within 10 meters—violates basic hygiene principles, raising the risk of environmental contamination through airborne particles or improper hand hygiene. Best practices in food safety recommend adequate physical separation between sanitation facilities and food service areas to minimize such risks.

In total, 76% of restaurants tested positive for at least one of the major foodborne pathogens. This rate is alarming when compared to similar studies: in Senegal [72], 14.3% of restaurants and 40.4% of meat carcasses tested positive for *Salmonella*, and in London [73], 88% of *Shigella* outbreaks were traced back to contaminated food served in restaurants. In the United States, raw ground beef was linked to 41% of *E. coli* outbreaks, underscoring the consequences of poor hygiene and temperature control.

The observed resistance patterns may also reflect antimicrobial selection pressure in livestock production systems. In many low- and middle-income countries, antimicrobials are frequently used for therapeutic, prophylactic, or growth pro-motion purposes in food-producing animals. Such practices can facilitate the emergence and spread of resistant bacteria within animal populations, which may subsequently enter the human food chain. Previous studies have demonstrated that antimicrobial use in livestock production is a major driver of antimicrobial resistance in foodborne pathogens.

AMR further compounds the public health challenge. Alarmingly, all isolates of *Salmonella*, *Shigella*, and *E. coli* in this study were 100% resistant to amoxicillin, a commonly used broad-spectrum antibiotic. *Salmonella* also exhibited complete resistance to erythromycin, a finding consistent with reports from India [74] and Nepal [75]. Resistance to doxycycline was found in 80% of *Salmonella*, 70% of *Shigella*, and 58% of *E. coli* O157:H7 isolates, while resistance to erythromycin was even higher: 92% for *Shigella* and 89.5% for *E. coli* O157:H7. These data indicate widespread multidrug resistance (MDR), especially to commonly prescribed antibiotics such as ciprofloxacin, erythromycin, and streptomycin. Only chlor-amphenicol showed retained effectiveness in this study, though 80% resistance to it was observed among *Shigella* iso-lates in Nepal [75]. Similar MDR trends have been reported in Mekelle, Ethiopia [76], where foodborne bacterial isolates from animal-origin foods demonstrated resistance to multiple antibiotics.

It is important to note that the high resistance observed for erythromycin may partly reflect intrinsic resistance mech-anisms in Gram-negative bacteria rather than solely acquired resistance. Enterobacterales possess outer membrane permeability barriers and efflux pumps that reduce susceptibility to macrolides. Nevertheless, monitoring such resistance patterns remains valuable for antimicrobial resistance surveillance and risk assessment.

Taken together, the high contamination rates, inadequate hygiene infrastructure, and extensive antibiotic resistance observed in this study represent a compounded health hazard. Raw beef consumers are therefore at increased risk not only of infection but also of treatment failure due to resistant pathogens. This calls for prompt interventions including public education on safe meat handling, strict regulatory enforcement, improved sanitary infrastructure in restaurants, and enhanced surveillance of antimicrobial resistance in foodborne pathogens.

## Limitation of the study

This study has several limitations. First, pathogen identification relied solely on culture and biochemical methods without molecular or serological confirmation, which may have resulted in misclassification, particularly between *Shigella* species and *Escherichia coli*, including *E. coli* O157:H7. Second, the risk scoring framework was observa-tional and semi-quantitative and lacked formal validation or probabilistic modelling. The thresholds used to catego-rize low, medium, and high risk were not statistically derived and should therefore be interpreted as indicative rather than predictive.

Third, some antibiotics included in the susceptibility testing panel, particularly erythromycin and amoxicillin, are not routinely used for Gram-negative infections but were retained to support antimicrobial resistance surveillance and com-parative epidemiological assessment in the local context. Because internationally validated clinical breakpoints are not uniformly available for all organism–drug combinations in Enterobacterales, these results should be interpreted cautiously

and not considered direct therapeutic guidance. To avoid potential overestimation of MDR, its classification should ideally rely on antimicrobial agents with validated interpretive criteria for the tested organisms.

Fourth, the statistical analysis was primarily descriptive, and formal hypothesis testing was limited. Finally, although the study adopts a One Health perspective, the investigation focused mainly on the foodborne human pathway and did not incorporate environmental or animal reservoir data. Future research should address these gaps through molecular confirmation methods, integrated One Health surveillance, and more robust statistical and risk modelling approaches.

## Conclusion

This study reveals a pressing public health concern linked to the consumption of raw beef in Ethiopian restaurants, marked by the high prevalence of *E. coli O157:H7*, *Shigella*, and *Salmonella*, and their alarming levels of multidrug resistance. The complete resistance of these pathogens to widely used antibiotics like amoxicillin underscores the risk of treatment failure and signals a deeper systemic gap in food safety management. Poor hygiene practices, lack of infrastructure, cultural consumption patterns, and weak regulatory enforcement create an environment conducive to bacterial contamination and AMR transmission. These challenges demand urgent, multisectoral action within a One Health framework, combining regulatory oversight, infrastructure improvement, food handler training, and public education. Protecting public health requires more than technical solutions, which calls for coordinated policy, community engagement, and a shared commitment to safe food across sectors.

## Supporting information

**S1 Text. S1 Excel.** *General characteristics of samples collected from five towns in Ethiopia.* This file presents metadata on the location of sampling sites, types of food-contact surfaces (e.g., knives, tables, meat surfaces, handlers' hands), and the number and type of bacterial isolates recovered from each site. It provides contextual information for interpreting microbial contamination patterns across different environments. **S2 Excel.** *Antimicrobial resistance profiles of bacterial isolates.* This data details the inhibition zone measurements, resistance interpretation categories (sensitive, intermediate, resistant), and data that was used to compute mean, standard deviation, and standard error values for each bacterial species against six antibiotics. These data complement the AMR prevalence summaries presented in the main manuscript and support the analysis of multidrug resistance patterns.
(ZIP)

## Author contributions

**Conceptualization:** Daniel Teshome Gebeyehu, Temechew Munaw Abebe, Gashaw Enbiyale Kasse.

**Data curation:** Daniel Teshome Gebeyehu.

**Formal analysis:** Daniel Teshome Gebeyehu, Temechew Munaw Abebe, Md Shahidul Islam.

**Funding acquisition:** Daniel Teshome Gebeyehu.

**Investigation:** Daniel Teshome Gebeyehu.

**Methodology:** Temechew Munaw Abebe, Ayalew Negash Shiferaw.

**Project administration:** Daniel Teshome Gebeyehu.

**Resources:** Daniel Teshome Gebeyehu.

**Software:** Temechew Munaw Abebe, Ayalew Negash Shiferaw, Gashaw Enbiyale Kasse.

**Supervision:** Ayalew Negash Shiferaw, Md Shahidul Islam, Gashaw Enbiyale Kasse.

**Validation:** Temechew Munaw Abebe, Ayalew Negash Shiferaw, Md Shahidul Islam, Gashaw Enbiyale Kasse.

**Visualization:** Temechew Munaw Abebe, Ayalew Negash Shiferaw, Md Shahidul Islam, Gashaw Enbiyale Kasse.

**Writing – original draft:** Daniel Teshome Gebeyehu.

**Writing – review & editing:** Daniel Teshome Gebeyehu, Temechew Munaw Abebe, Ayalew Negash Shiferaw, Md Shahidul Islam, Gashaw Enbiyale Kasse.

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
