## [Decision Letter · Decision Letter 0]

6 Feb 2026

Dear Dr. Gebeyehu,

plosone@plos.org. . . . A letter that responds to each point raised by the academic editor and reviewer(s). You should upload this letter as a separate file labeled 'Response to Reviewers'.A marked-up copy of your manuscript that highlights changes made to the original version. You should upload this as a separate file labeled 'Revised Manuscript with Track Changes'.An unmarked version of your revised paper without tracked changes. You should upload this as a separate file labeled 'Manuscript'.

We look forward to receiving your revised manuscript.

Kind regards,

Guadalupe Virginia Nevárez-Moorillón, Ph.D.

Academic Editor

PLOS One

Journal Requirements:

“Wollo University funded the data collection and laboratory investigation expenses of this research project, and the funder has no competing interest and has no role in publishing this paper.”

“Wollo University funded the data collection and laboratory investigation expenses of this research project.”

“Wollo University funded the data collection and laboratory investigation expenses of this research project.”

“Wollo University funded the data collection and laboratory investigation expenses of this research project.”

We note that one or more of the authors is affiliated with the funding organization, indicating the funder may have had some role in the design, data collection, analysis or preparation of your manuscript for publication; in other words, the funder played an indirect role through the participation of the co-authors. If the funding organization did not play a role in the study design, data collection and analysis, decision to publish, or preparation of the manuscript and only provided financial support in the form of authors' salaries and/or research materials, please do the following:

1. Review your statements relating to the author contributions, and ensure you have specifically and accurately indicated the role(s) that these authors had in your study. These amendments should be made in the online form.

2. Confirm in your cover letter that you agree with the following statement, and we will change the online submission form on your behalf:

“The funder provided support in the form of salaries for authors [insert relevant initials], but did not have any additional role in the study design, data collection and analysis, decision to publish, or preparation of the manuscript. The specific roles of these authors are articulated in the ‘author contributions’ section.

6. We note that your Data Availability Statement is currently as follows: “All data generated or analysed during this study are included in this published article and its supplementary information files.”

Reviewers' comments:

Reviewer's Responses to Questions

**Comments to the Author**

1. Is the manuscript technically sound, and do the data support the conclusions?

Reviewer #1: Partly

Reviewer #2: Partly

Reviewer #3: Yes

2. Has the statistical analysis been performed appropriately and rigorously?

Reviewer #1: Yes

Reviewer #2: No

Reviewer #3: No

3. Have the authors made all data underlying the findings in their manuscript fully available?

Reviewer #1: Yes

Reviewer #2: Yes

Reviewer #3: Yes

4. Is the manuscript presented in an intelligible fashion and written in standard English?

Reviewer #1: Yes

Reviewer #2: No

Reviewer #3: Yes

Reviewer #1: The research is interesting and raises serious concerns about the potential public health risks associated with the consumption of raw beef in Ethiopia. I support its publication after appropriate modifications, as outlined below.

The names of bacterial species should be written in italics. There are some omissions, so please check carefully and correct them.

In microbiological food analysis, standardized methods require the use of control strains to ensure the validity of the results. Hence, it is necessary to specify which bacterial strains were used as controls (positive control and negative control, respectively).

Most antibiotics tested to assess the susceptibility of isolated strains, except ciprofloxacin, are not currently used to treat infections with E. coli, Salmonella, or Shigella due to recognized antimicrobial resistance, toxicity, and the availability of superior alternatives. Therefore, the authors should explain the criteria used to select the antimicrobials for this study.

The presence of E. coli strains in raw beef is not unusual, and not all strains of E. coli are potentially pathogenic (with the exception of E. coli O157:H7 or other STEC). In fact, E. coli is an indicator of hygiene in food industry, whose presence is allowed in a limited count, depending on the legislation or reference standard. Therefore, the authors should reconsider these statements in the discussion section.

In Conclusions section the authors need to insert a complex sentence highlighting the study limitation and further perspectives in the approached research area.

Reviewer #2: This manuscript by Gebeyehu et al. addresses an important and timely public health issue: antimicrobial-resistant foodborne pathogens associated with raw beef consumption in Ethiopia, framed within a One Health perspective. The topic is relevant, and the dataset appears substantial. However, In its present form, the manuscript does not yet meet the standards for publication in a high-impact, international journal. I invite the authors to address the following concerns, specifically:

The manuscript claims to conduct a “health risk assessment,” yet the approach used is largely descriptive and semi-quantitative, without a clearly defined or validated risk assessment model. The risk scoring system (likelihood × severity) is insufficiently justified, not formally validated, and relies heavily on subjective scoring of hygiene practices. The manuscript does not clearly articulate what is novel relative to prior Ethiopian studies on raw meat contamination and AMR. Several similar prevalence studies are cited, but the incremental contribution of this work remains unclear. Therefore, the authors must clearly define whether this study is a microbiological prevalence study, an AMR surveillance study, or a formal risk assessment. If it is a risk assessment, the authors must justify the framework, explain assumptions, and clarify how it differs from existing Codex/WHO quantitative or semi-quantitative approaches. In addition, they must explicitly state the novel contribution in the Introduction and Discussion sections.

Identification of Salmonella, Shigella, and E. coli O157:H7 relies entirely on culture and biochemical methods, without molecular confirmation.Given the high reported prevalence of Shigella and E. coli O157:H7, the absence of PCR or serological confirmation raises concerns about misclassification. Shigella is rarely isolated from meat compared to Salmonella and E. coli; the unusually high prevalence requires stronger validation. Therefore, the authors must acknowledge the limitations of culture-based identification explicitly and clarify whether any serotyping or molecular confirmation was attempted. Likewise, temper claims related to E. coli O157:H7 and Shigella prevalence, or justify them with stronger methodological support.

Within the antimicrobial susceptibility testing section, the antibiotic panel includes drugs of questionable relevance for Gram-negative enteric pathogens (e.g., erythromycin). Chloramphenicol is reported as 100% effective, yet this drug is largely obsolete or restricted in many settings due to toxicity. CLSI breakpoints for some antibiotics used (especially erythromycin) are not clearly applicable to the organisms tested. The authors must justify the choice of antibiotics based on national treatment guidelines or surveillance relevance and clarify CLSI breakpoint applicability for each organism–antibiotic combination. What about chloramphenicol results and their interpretation? Avoid presenting it as a practical treatment option.

The hygiene scoring system (binary compliant/non-compliant) lacks validation and weighting. The reviewer wonder, why? The calculation of “cumulative gaps” (e.g., 52%, 76%, 100%) is arbitrary, with no sensitivity analysis. Risk categorization (low/medium/high) is asserted rather than statistically or probabilistically demonstrated. Please clarify these concerns, by providing a clear mathematical description of how risk scores were derived and justify thresholds for “high risk” using published frameworks.

Statistical methods are largely descriptive. Associations between hygiene practices and contamination are mentioned, but results of statistical tests are not presented (e.g., test statistics, p-values). Please clarify this issues!

The manuscript frequently uses strong causal language (“poses a serious health threat,” “urgent need”) without sufficient causal evidence. The One Health framing is conceptually invoked but not operationalized through data from animal health, environmental sampling, or policy analysis.

Last, but not least, language and grammar require substantial professional editing.

Reviewer #3: The manuscript discusses the public health risks of consuming raw beef in Ethiopia, especially in relation to bacterial contamination and antimicrobial resistance (AMR). The topic is highly relevant within the One Health concept, linking food safety, antimicrobial resistance, and consumer health. The manuscript provides useful primary microbiological data, hygiene observations, and risk assessment, making it potentially valuable for policymakers and public health researchers. Overall, this manuscript is well-written but needs some revisions and explanation. I recommend this manuscript for a major revision. I explain my concerns in more detail below. I ask that the authors specifically address each of my comments in their responses.

Comments:

1. Introduction:

• Why in this manuscript E. coli, Salmonella and Shigella were prioritized? Please add deep explanation or references.

• Research gap could be sharpened

• The introduction would benefit if the Authors can give clearer explanation about how impact the AMR to raw beef meat consumers.

2. Material and methods:

• Please add how to select 50 restaurant and 250 swab samples.

• Study design: The study is use cross-sectional, yet sampling occurred over a six-month period (January–June 2023). Clarification is needed on whether samples were collected once per site or repeatedly, and how temporal variability was addressed.

• 2.5.1 and 2.5.2 The authors should add molecular data like PCR, etc to confirm the bacteria and MDR.

• Could the Authors explain, why for antibiotic resistance testing use 6 antibiotics? Please give the additional explanation.

• Why the Authors use Erythromycin and Amoxicillin? Because there are not relevant to the Gram-negative bacteria.

• Data analysis: Criteria for significance are not mentioned and the rationale for using Chi square & Fisher’s test should be briefly stated.

3. Results:

• Explanation in 3.3 more appropriate for the discussion

• The criteria used to define multidrug resistance (MDR) should be explicitly stated in the Results section before MDR percentages are reported.

• The Results section would be strengthened by presenting confidence intervals or measures of uncertainty for key prevalence estimates.

• Although the Methods section mentions chi-square and Fisher’s exact tests, the Results does not report corresponding test statistics, degrees of freedom, or p-values. Where associations are implied (e.g., between hygiene practices and contamination), statistical outcomes should be reported.

4. Discussion:

• Deeper analytical interpretation rather than descriptive comparison

• Insufficient acknowledgment of study limitations

• Please be careful with the statement of treatment failure or severe clinical outcomes because this study does not provide direct clinical evidence of infection, therapeutic failure, or adverse health outcomes. The data only from meat and restaurant environments.

5. Conclusion:

• Please make it simply, the conclusion is too long. Conclude the result from your manuscript.

• Add future research in this section.

.

Reviewer #1: No

Reviewer #2: No

Reviewer #3: **Yes:**Dhandy Koesoemo WardhanaDhandy Koesoemo WardhanaDhandy Koesoemo WardhanaDhandy Koesoemo Wardhana

---

## [Author Response · Author response to Decision Letter 1]

16 Feb 2026

Reviewer comments led to substantial methodological, conceptual, and linguistic improvements. Scientific nomenclature was standardized, including italicization of bacterial species. Control strains were specified, and antibiotic selection criteria were justified based on local clinical relevance and surveillance value. The definition of multidrug resistance (MDR) was explicitly stated. Sampling procedures, study design, statistical methods, and criteria for significance were clarified. The statistical analysis section was revised, and limitations related to subgroup size and power were transparently acknowledged.

The study framework was refined and clearly defined as a semi-quantitative microbial risk estimation rather than a formal quantitative risk assessment. The risk scoring model, assumptions, and rationale were clarified. Concerns regarding culture-based identification without molecular confirmation were explicitly acknowledged, and claims regarding pathogen prevalence were appropriately tempered. Causal language throughout the manuscript was revised to reflect association rather than inference of clinical outcomes.

The research gap, novelty, and One Health framing were strengthened in the Introduction and Discussion. A dedicated limitations section was added, the discussion was deepened analytically, and the conclusion was condensed with clear future research directions.

Overall, the manuscript has been substantially revised to address all editorial and reviewer concerns.

A file named "response to reviewers" is uploaded.

---

## [Decision Letter · Decision Letter 1]

2 Mar 2026

Dear Dr. Gebeyehu,

We look forward to receiving your revised manuscript.

Kind regards,

Guadalupe Virginia Nevárez-Moorillón, Ph.D.

Academic Editor

PLOS One

Journal Requirements:

Additional Editor Comments:

Please address the comments of Reviewer 3

Reviewer's Responses to Questions

**Comments to the Author**

Reviewer #1: (No Response)

Reviewer #2: All comments have been addressed

Reviewer #3: (No Response)

2. Is the manuscript technically sound, and do the data support the conclusions?

Reviewer #1: Yes

Reviewer #2: Yes

Reviewer #3: Partly

3. Has the statistical analysis been performed appropriately and rigorously?

Reviewer #1: Yes

Reviewer #2: Yes

Reviewer #3: Yes

4. Have the authors made all data underlying the findings in their manuscript fully available?

Reviewer #1: (No Response)

Reviewer #2: Yes

Reviewer #3: Yes

5. Is the manuscript presented in an intelligible fashion and written in standard English?

Reviewer #1: Yes

Reviewer #2: Yes

Reviewer #3: Yes

Reviewer #1: (No Response)

Reviewer #2: The authors correctly acknowledged all of my raised concenrs in the previous review round. Congratulation!

Reviewer #3: I would like to thank the authors for their efforts in revising the manuscript and for addressing many of the previous comments. However, there remain some important issues that need further clarification and refinement.

1. The inclusion of erythromycin and amoxicillin in the antimicrobial susceptibility testing requires further clarification. Although the authors justify their selection based on local antimicrobial usage and surveillance value, previous studies consistently demonstrate that these antibiotics are generally ineffective against Gram-negative bacteria such as E. coli, Salmonella, and Shigella due to intrinsic resistance. Therefore, the scientific rationale for testing these agents should be more clearly articulated. If resistance is expected to approach 100% based on well-established intrinsic mechanisms, the purpose and added value of reporting these resistance rates need to be explicitly explained.

The authors should also clearly state whether antimicrobial susceptibility testing was performed and interpreted according to internationally recognized standards, such as the Clinical and Laboratory Standards Institute (CLSI) or the European Committee on Antimicrobial Susceptibility Testing (EUCAST), and specify the breakpoints applied for each organism–antibiotic combination.

If the inclusion of these antibiotics was intended to reflect local antimicrobial usage patterns (e.g., in veterinary practice), the manuscript should provide supporting data or appropriate references documenting such usage. Furthermore, it should be clarified whether intrinsically resistant agents were excluded from multidrug resistance (MDR) calculations to avoid potential overestimation of resistance levels.

Without these clarifications, the interpretation, scientific validity, and epidemiological relevance of the susceptibility data remain uncertain.

2. Discussion part

The Discussion section is currently largely descriptive and comparative in nature. Many of the interpretation follows the pattern of reporting prevalence values and comparing them with findings from other countries, followed by generalized explanations such as differences in hygiene practices. While such comparisons are useful, the section lacks sufficient analytical depth expected for publication in an international peer-reviewed journal.

The authors can give deep explanation about the results, for example:

- What biological or ecological factors could explain the unusually high prevalence?

- Could supply chain conditions contribute to the contamination?

- Does this suggest potential failure in cold chain management during transportation or storage?

- Whether the reported resistance (particularly to amoxicillin and erythromycin) reflects intrinsic resistance rather than acquired resistance.

- Whether there is evidence of antibiotic selection pressure in livestock production systems.

- The possible role of unregulated or excessive antimicrobial use in veterinary practice.

and many additional aspects that could be discussed based on your findings.

.

Reviewer #1: No

Reviewer #2: No

Reviewer #3: No

---

## [Author Response · Author response to Decision Letter 2]

17 Mar 2026

Response to reviewer letter is attached.

---

## [Editor Report · Decision Letter 2]

22 Mar 2026

Antimicrobial-resistant pathogens on the plate: a semi-quantitative hygiene risk evaluation of raw beef consumption in Ethiopia within a One Health context

PONE-D-25-56353R2

Dear Dr. Gebeyehu,

We’re pleased to inform you that your manuscript has been judged scientifically suitable for publication and will be formally accepted for publication once it meets all outstanding technical requirements.

Kind regards,

Guadalupe Virginia Nevárez-Moorillón, Ph.D.

Academic Editor

PLOS One
---

## [Editor Report · Acceptance letter]

PONE-D-25-56353R2

PLOS One

Dear Dr. Gebeyehu,

I'm pleased to inform you that your manuscript has been deemed suitable for publication in PLOS One. Congratulations! Your manuscript is now being handed over to our production team.

Kind regards,

on behalf of

Dr. Guadalupe Virginia Nevárez-Moorillón

Academic Editor

PLOS One